# High-precision electron affinity of oxygen

Moa K. Kristiansson [1] ✉, Kiattichart Chartkunchand [1,2], Gustav Eklund [1], Odd M. Hole[1], Emma K. Anderson[3], Nathalie de Ruette[1], Magdalena Kamińska[1], Najeeb Punnakayathil [1], José E. Navarro-Navarrete[1], Stefan Sigurdsson[1], Jon Grumer [4], Ansgar Simonsson [1], Mikael Björkhage[1], Stefan Rosén[1], Peter Reinhed[1], Mikael Blom[1], Anders Källberg[1], John D. Alexander[1], Henrik Cederquist [1], Henning Zettergren [1], Henning T. Schmidt [1] & Dag Hanstorp[5]

Negative ions are important in many areas of science and technology, e.g., in interstellar chemistry, for accelerator-based radionuclide dating, and in antimatter research. They are unique quantum systems where electron-correlation effects govern their properties. Atomic anions are loosely bound systems, which with very few exceptions lack optically allowed transitions. This limits prospects for high-resolution spectroscopy, and related negative-ion detection methods. Here, we present a method to measure negative ion binding energies with an order of magnitude higher precision than what has been possible before. By laser-manipulation of quantum-state populations, we are able to strongly reduce the background from photodetachment of excited states using a cryogenic electrostatic ion-beam storage ring where keV ion beams can circulate for up to hours. The method is applicable to negative ions in general and here we report an electron affinity of 1.461 112 972(87) eV for $^{16}$O.

Negative ions are essential in many natural environments and in a large range of applications. Recent observations of molecular negative ions in space have led to a boost in the field of their spectroscopy, e.g., refs. 1–3. Negative ions are present in many types of plasmas where they influence the formation and destruction of molecules and act as important charge carriers e.g., in interstellar clouds[1,4]. Their unique properties are important in Accelerator Mass Spectrometry; the most sensitive trace-element detection method, with [14]C-dating as the best-known example[5]. Other research areas gaining much focus recently are laser cooling of negative ions, which would allow for sympathetic cooling of anti-protons[6–8], and photo-detachment of the negative ion of positronium, Ps⁻, useful for creating energy-tunable beams of neutral Ps to investigate collisions between Ps and regular matter[9]. Furthermore, the first electron affinity measurement of a radioactive element was recently conducted at the isotope facility ISOLDE at CERN, where the negative ion of astatine, which is of interest for targeted radiotherapy[10], was investigated[11]. Many chemical properties of an element, e.g., the

electronegativity or the electrophilicity index, can be determined from its electron affinity and ionization energies[11–14]. Therefore, high-precision measurements of these entities are important.

Atomic negative ions differ from neutral atoms and positive ions in which long-range forces are dominating. The attractive Coulomb potential felt by a given valence electron in an atomic negative ion is screened by a number of other electrons equal to the proton number. These valence electrons thus experience little or no long-range net force. The binding of the ion is instead made possible by a $1/r^4$ polarization potential[15]. Hence, electron correlation plays an essential role in the properties of negative ions, and for many elements in the periodic table, bound-state Hartree-Fock wavefunctions do not exist, not even for the pure two-electron system H⁻ which is, in fact, bound by 0.754,195(19) eV[16]. A model including correlation effects is needed to perform even rough predictions of bound-state properties and electron affinities[17]. Negative atomic ions thus constitute ideal systems to benchmark atomic theories going beyond the independent-particle model.

[1]Department of Physics, Stockholm University, Stockholm, Sweden. [2]Atomic, Molecular and Optical Physics Laboratory, RIKEN, Saitama, Japan. [3]Department of Physics and Astronomy, Aarhus University, Aarhus, Denmark. [4]Theoretical Astrophysics, Department of Physics and Astronomy, Uppsala University, Uppsala, Sweden. [5]Department of Physics, University of Gothenburg, Gothenburg, Sweden. ✉e-mail: moa.kristiansson@fysik.su.se

As a consequence of the short-range polarization potential, electron affinities are typically an order of magnitude smaller than the first ionization energies. Further, negative ions typically have only a few bound excited states, and in almost all cases, they have the same parity as the ground state, making traditional spectroscopy utilizing electric-dipole transitions impossible. Only five atomic negative ions with an excited state with a parity opposite to the ground state have been found[7,8,18–21]. The most characteristic general property of negative ions is the electron affinity of the corresponding neutral atom or molecule. This quantity can be measured through the photodetachment process, where the valence electron is emitted due to the absorption of a photon with known energy. The most precise measurement of an electron affinity to date is that of sulfur, where the electron affinity was determined with a 0.6 μeV experimental uncertainty (corresponding to a laser frequency uncertainty of 145 MHz) using photodetachment microscopy[22].

In recent years, the field of negative ions spectroscopy has been making use of the rapid development of cryogenic electrostatic ion-storage rings[23–26]. The cold environment and good vacuum provide very long storage times for atomic and molecular ions. For negative ions, these storage rings have been used to measure the lifetimes of long-lived metastable excited states[27–29]. In this work, we present a method for precision measurements of electron affinities where we conduct the photodetachment spectroscopy in the Double Electrostatic Ion Ring ExpEriment (DESIREE)[23,24]. Previous experiments using laser photodetachment threshold spectroscopy have been limited by a severe background due to photodetachment from excited states in the negative ions. The long storage times in the cryogenic ring allow for photodetachment of the excited ions using a high-power laser, leaving only ground-state ions in the ring. The photon energy is then scanned over the threshold region at a lower laser power to make depletion of the beam during the scan insignificant.

In this work, we demonstrate a method for high-precision photodetachment of negative ions by the measurement of the electron affinity of oxygen. A low background below the photodetachment threshold in combination with a narrow-linewidth laser and high-precision wavelength meter results in the most accurate measurement of an electron affinity so far, yielding a result of 1.461,112,972(87) eV for oxygen. This method can be used to measure electron affinities of any atomic element or molecule that form bound states when an additional electron is attached. Increased precision in the electron affinity will allow for more critical investigations of electron correlation

effects e.g., through high-precision experiments in isotope shifts of electron affinities.

## Results
### Experimental procedure

The most general method to measure electron affinities and negative ion excited state binding energies is laser photodetachment threshold spectroscopy, where a laser beam is overlapped with an ion beam. The photon energy is scanned near the photodetachment threshold and the rate of neutral atoms resulting from photodetachment is measured as a function of photon energy. The photodetachment cross-section in a narrow range above the electron affinity threshold can be described by the Wigner threshold law

$$\sigma_{th} \propto (E - E_{EA})^{l+1/2}, \tag{1}$$

where $E$ is the photon energy, $E_{EA}$ is the electron affinity, and $l$ is the angular momentum quantum number of the outgoing electron[30]. In the case of oxygen, the 2p electron can be emitted as an $s$ or $d$ electron. Close to the threshold, the $s$-wave dominates due to the centrifugal barrier for the $d$-wave, yielding a sharp onset ($l=0$ in Eq. (1) for the $s$-wave) that can be determined with high precision[31,32]. However, the upper fine-structure level of O⁻ has a lifetime of several hours[33]; any population in this level will thus normally cause a photodetachment signal below the electron affinity threshold, giving rise to a large background.

Negative ion production using ion sources with ionization rates high enough to produce usable beam currents for photodetachment threshold experiments (typically from 100 pA to a few nA) leads to large populations of all exciting levels since the energies involved in ion production are in general much larger than the energy splittings between the involved levels[34,35]. Therefore, the distribution of ions populating the ground state and excited states does not typically follow a Boltzmann distribution but is instead determined by the degeneracy of the fine-structure levels, leaving a large fraction of the ions in excited states. Excited states will often give a large contribution to any background signal below the photodetachment threshold. Thus, an ion beam with essentially all ions in the ground state is needed for high-precision electron affinity measurements.

By waiting for ions to equilibrate with the surrounding black-body radiation, ground-state populations exceeding 90% were demonstrated when beams of OH⁻ were stored in cryogenic storage rings[36,37]. When spontaneous processes are too slow, one can apply selective destruction by laser manipulation to control the quantum-state distribution of the ion beam[38]. By applying selective laser photodetachment, the long-lived rotationally excited states of the OH⁻ ion were successfully depleted, further increasing the ground-state population for an ion beam stored in an electrostatic cryogenic storage ring[37]. A similar technique was used for atomic negative ions in a recent study by Müll et al., where an ion beam purely in the $^4S_{3/2}$ ground state of Si⁻ was produced by selectively photodetaching excited state ions[29].

The excited metastable state of the oxygen anion, $^2P_{1/2}$, is a fine-structure level that belongs to the same term as the $^2P_{3/2}$ ground state, lying about 0.022 eV above the ground state as shown in Fig. 1. In many of the previous photodetachment experiments on O⁻, e.g., in refs. 22, 39–42, the excited state is highly populated. The current experiment utilizes a storage ring held at 13 K and a high-power laser to produce an ion beam almost completely in the ground state, thus avoiding photodetachment from the excited state that would otherwise affect the measurements. The experiment is performed at the DESIREE facility operated at 13 K and using one of its two storage rings[23,24]. A schematic of the storage ring is shown in Fig. 2.

During a single measurement cycle, O⁻ ions are stored for 165 s and probed by laser beams either co- or counter-propagating with respect to the ion beam. During the first 35 s, the ions in the excited

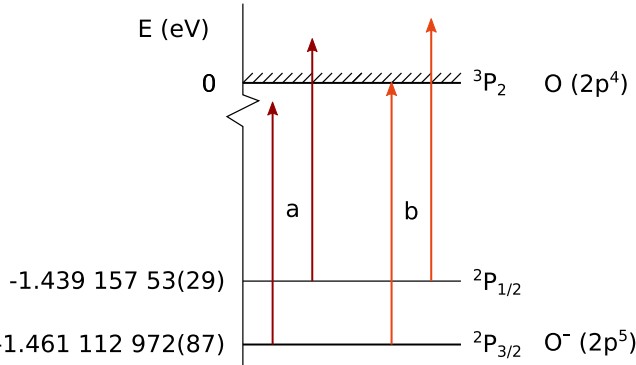

**Fig. 1 | Energy levels of O⁻.** The ground state has the electron configuration $1s^2 2s^2 2p^5$ $^2P_{3/2}$. We measure binding energies of 1.461,112,972(87) eV and 1.43,915,753(29) eV for the $^2P_{3/2}$ and $^2P_{1/2}$ levels, respectively. The photon energy arrows labeled "a" represent the photon energy needed for photodetachment from the excited level only, whereas the arrows labeled "b" represent the photon energy needed to photodetach from both the ground state and excited state. The hatched area marks the continuum of an oxygen atom in its ground state and a free electron.

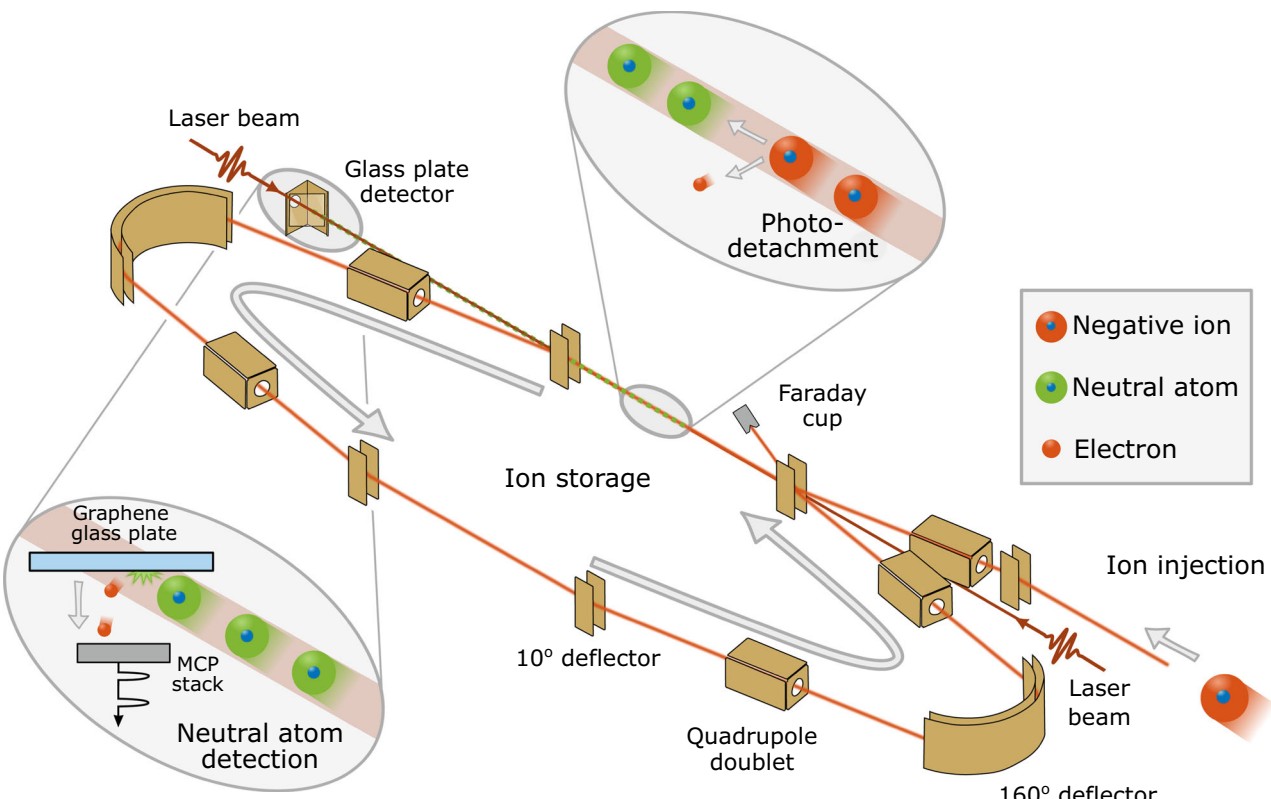

**Fig. 2 | A schematic of the experimental setup.** Ions are injected into the storage ring and photodetached using a continuous laser in a parallel alignment along the straight section of the injection line. The laser can be applied in either co-propagating or counter-propagating directions. The neutral atoms created in the photodetachment process are detected by a detector system located after the straight section. The detector system consists of a glass plate covered with a layer of graphene which emits slow secondary electrons when hit by neutral oxygen atoms. The secondary electrons are amplified and detected by using a stack of microchannel plates (MCP). The ions are guided by bending deflectors and focused by quadrupole doublets in the storage ring.

state are selectively photodetached using 2.5 W of laser power, almost completely emptying the ion beam of ions in the excited state. Thereafter, the laser power is reduced, and, using the same ion beam, the photon energy is scanned over the electron affinity threshold four times, while the neutrals created from photodetachment are measured as a function of photon energy. The ions are then dumped and a new measurement cycle is started with the direction of the laser beam reversed. The four scans from each measurement cycle are individually binned with bin sizes varying between 300–500 MHz. The detected number of events in each bin is assumed to follow a Poisson distribution and the uncertainties (one standard deviation) are thus calculated as the square root of the number of counts in each bin. The data is fitted using a convolution between the Wigner law and a Gaussian distribution representing the energy spread of the ion beam. The Wigner law with $l = 0$ is expressed as

$$\sigma(E) = A(\sqrt{E - E_{th}})\theta(E, E_{th}) + C, \tag{2}$$

where $A$ and $C$ are constants, $E$ is the photon energy, $E_{th}$ is the threshold energy, and $\theta$ is a Heaviside step function defined as 1 above the threshold and 0 below the threshold. The constant $C$ is the level of the signal below the threshold and reducing its value is crucial to accurately determine the threshold value. The Gaussian is expressed as

$$g(E) = \frac{1}{\sigma\sqrt{2\pi}}\exp\left(\frac{-E^2}{2\sigma^2}\right). \tag{3}$$

Here, $2\sqrt{2\ln 2}\sigma$ is the full width at half maximum of the energy distribution, $g(E)$.

## Precision measurements

Examples of two threshold scans, with the laser beam parallel and anti-parallel to the ion beam, are shown in Fig. 3, together with the threshold fit as described above. The signal below the threshold (the level of which corresponds to the constant, $C$, in Eq. (2)) is a combination of detector background, signal from atoms created by collisions with the residual gas, and a small contribution from photodetachment of ions remaining in the excited state. Without depletion of ions in the excited state, the background signal would have been more than 20 times larger, and would thus have prevented an accurate threshold determination. The two threshold values, $E_{EA}^{p}$ for parallel alignment and $E_{EA}^{a}$ for anti-parallel alignment, are combined as the geometric mean since this gives the Doppler-free electron affinity, $E_{EA}$ according to ref. 43:

$$\sqrt{E_{EA}^{p}E_{EA}^{a}} = \sqrt{\frac{1+v/c}{\sqrt{1-v^2/c^2}}E_{EA}\frac{1-v/c}{\sqrt{1-v^2/c^2}}E_{EA}} = E_{EA} \tag{4}$$

The results from the first scan in a parallel alignment measurement is averaged with the results from the first scan in an anti-parallel alignment measurement, etc. From one set of measurements, four Doppler-free threshold values are obtained using Eqn. (4). The procedure is repeated a large number of times for different laser powers. The latter is to investigate how the laser affects the ion-beam energy distribution and, possibly also, as a consequence of this, the measured threshold position. The final electron affinity value is obtained from measurements where a sufficiently low laser power was used to avoid affecting the threshold position. The total number of scans included in the final analysis is 228 independent measurements of the electron affinity, as illustrated in Fig. 4. The weighted arithmetic mean of these

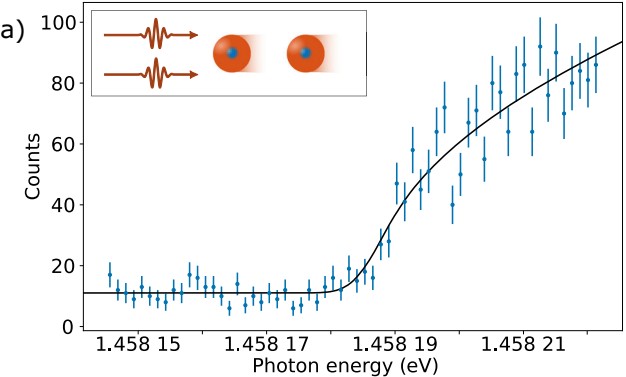

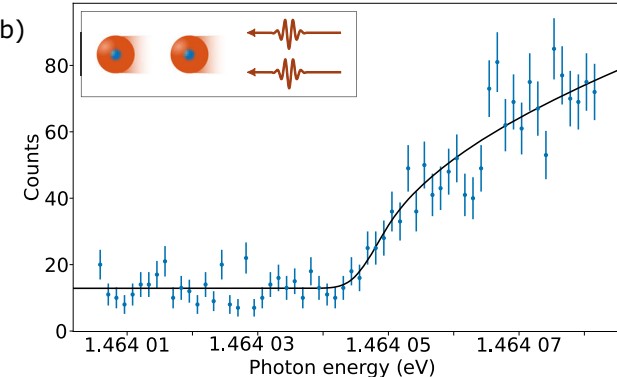

**Fig. 3 | Examples of two single threshold scans.** The laser propagates anti-parallel (**a**) or parallel (**b**) with respect to the ion beam. The data shown in blue are binned using 300 MHz (1.24 µeV) bins and a fit using a convolution between the Wigner law and a Gaussian distribution is shown in black. The error bars are the square root of the number of events in each bin. The threshold values are **a** 1.458,185,59(79) eV and **b** 1.464,046,29(95) eV, respectively. The geometrical mean of the two thresholds, as given by Eqn. (4), is 1.461,113,00(62) eV. The final electron affinity value is obtained as the weighted average of 228 combined thresholds such as this pair.

228 thresholds gives an electron affinity of 1.461,112,972(45) eV where the uncertainty, one standard deviation, corresponds to 11 MHz in the frequency domain.

## Sources of uncertainties

The final uncertainty of the electron affinity value includes the uncertainty in the photon energy measurement from the wavemeter and any systematic effects present in the experiment. The uncertainty in the measured photon energy is given by the manufacturer as 0.041 µeV (10 MHz). As mentioned above, we have investigated the possibility that the interaction with the laser light affects the velocity distribution of the circulating ions and hence the measured threshold value. We only used measurements with laser powers in a region where no power dependence was found. To be conservative, however, we add uncertainty of 0.062 µeV (15 MHz) as an upper limit. The statistical uncertainty of 0.045 µeV (11 MHz), together with the wavemeter and photodetachment uncertainties of 0.041 and 0.062 µeV (10 and 15 MHz), respectively, are added in quadrature to give a total uncertainty of 0.087 µeV (21 MHz). This results in an electron affinity of 1.461,112,972(87) eV.

## Fine-structure splitting

In addition to the electron affinity, the photodetachment threshold corresponding to the binding energy of the upper fine-structure level $^2P_{1/2}$ is also investigated. The number of ions in the excited level present in the ion beam is smaller than the number of ions in the ground state. This, in combination with a smaller photodetachment cross-section[42], gives a very small photodetachment signal close to the threshold. Therefore, a slightly higher laser power of 215 mW is used to obtain enough signal. The measurement procedure is simpler than for the ground state since no depletion with the high-power laser is needed. Instead, the photon energy scan is started 10 s into the storage cycle and three scans over the threshold are performed before changing to the other laser propagation direction. Ten (10) measurement sets, with three scans in each, are performed and the resulting geometric mean of the 30 scans gives a threshold value of 1.439,157,53(20) eV. An additional uncertainty of 0.21 µeV (50 MHz) is added due to the high laser power used while scanning. The resulting uncertainty is then 0.29 µeV (70 MHz). This gives an excited state binding energy of 1.439,157,53(29) eV. The fine structure splitting is the difference between this binding energy and the electron affinity, and the result is 0.021,955,44(30) eV.

## Discussion

In this work, we present a method for high-precision measurements of photodetachment threshold energies of negative ions. The method is demonstrated with a measurement of the electron affinity of oxygen, resulting in a statistical uncertainty of 11 MHz and a final electron affinity uncertainty of 21 MHz. The results are compared to previous experimental electron affinity measurements in Fig. 5. The most recent, previous measurements agree with our measured value of 1.461,112,972(87) eV[22,40]. However, the present result has a significantly reduced uncertainty. The value reported by ref. 39, obtained by the laser photodetachment threshold spectroscopy technique, differs by about 3.4 µeV, which is significantly more than expected from their stated uncertainty.

Negative ions have the remarkable property of being weakly bound systems largely dominated by electron correlation. Therefore, electron-affinity calculations represent one of the most challenging situations for atomic structure theory. For example, variational approaches such as configuration-interaction or multiconfigurational Hartree-Fock (MCHF) methods typically require convergence of the absolute energies of the involved atomic eigenstates, something which is rarely achievable in practice. Instead, one has to rely on the relative convergence between the neutral atom and the negative ion. Balancing the complex correlation model of the negative ion with that of the neutral atom poses a major theoretical undertaking and the model can easily converge to the wrong results. Several computations of the oxygen electron affinity, based on various theoretical approaches are available in the literature, e.g., refs. 44–48. The study by Godefroid and Fischer, based on the MCHF method with relativistic Breit–Pauli corrections, reports a value for the $^{16}$O electron affinity of 1.470,4 eV[47], i.e., deviating by 9.3 meV from our experimental value. The coupled-cluster method was applied by Klopper et al., yielding a value of 1.461,04 eV[48], i.e., with a deviation of 703 µeV from the present value. However, these calculations are more than 10 years old and, since then, much development regarding theoretical models and computing power has taken place.

In addition to the electron affinity, we also present a measured value for the fine-structure splitting in the $^2P$ ground term of 21.955,44(30) meV. Previous measurement of this resulted in 21.955,6(17) meV[49], in agreement with our value. In contrast to the electron affinity, this splitting is easier to predict theoretically since it is dominated by spin-orbit effects within the same term. This is illustrated by comparison to the calculation by Godefroid and Fischer, which resulted in a splitting of 22.110 meV[47]. Here, the discrepancy between theory and experiment is smaller and amounts to 0.155 meV. For the relatively simple case of oxygen, the theoretically predicted energy structure agrees reasonably well with the experimental results. This is, however, in contrast to many other cases of negative ions with more complex bound-state structures, e.g., refs. 18–20, 50.

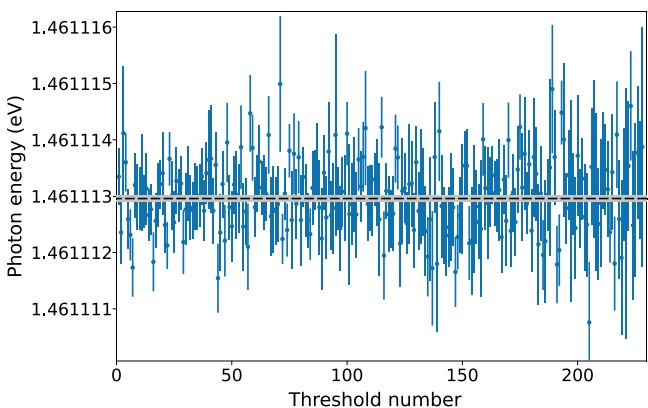

**Fig. 4 | All threshold values used for the final electron affinity.** The 228 individual threshold values used for the final electron affinity determination are in blue. The uncertainties obtained from the individual fits are given as the error bars. The weighted arithmetic mean of the thresholds gives an electron affinity of 1.461,112,972(45) eV for oxygen. The electron affinity is illustrated by the dashed black line and the statistical 0.045 μeV uncertainty is indicated by the gray area.

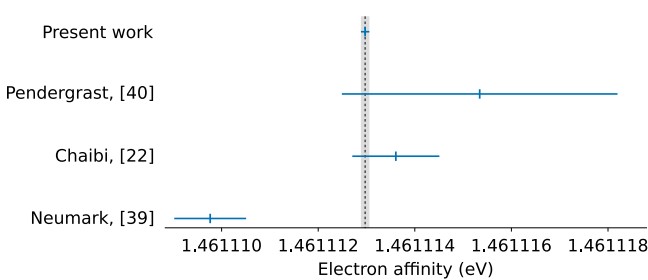

**Fig. 5 | Comparison with previous experimental results of the electron affinity, in eV, of $^{16}$O.** Our measured electron affinity is compared with a selection of the previous experimental results. The uncertainties are the ones given in each corresponding reference.

The situation, with regards to the theoretical accuracies, changes when the isotope shift of the electron affinity is investigated. Here, theoretical predictions are often closer to the experimentally measured values[47,51,52]. The isotope shift consists of the normal mass shift, the specific mass shift, and the field shift. The normal mass shift is easily calculated from the masses of the two isotopes. The specific mass shift, which is determined by the electron correlation, is much more complicated to calculate. The field shift is caused by the deviation of the 1/r potential as the electrons start to penetrate the nucleus and is also a computational challenge. Measuring this shift is the main method used to determine the charge distribution in nuclei. However, the only measurable quantity is the total isotope shift. Hence, investigations of the nuclear structure via measurements of atomic properties are dependent on accurate calculations of the specific mass shift.

Godefroid and Fischer have reported calculations of the isotope shift in the electron affinity of $^{18-16}$O (EA($^{18}$O)-EA($^{16}$O))[47]; the resulting value of −7.104 μeV agrees with the experimental value of −9.2(2.2) μeV obtained by ref. 49. With the present method, it is possible to determine electron affinities, and therefore isotope shifts on the electron affinity, with an absolute uncertainty on the order of 0.1 μeV, further challenging the theoretical calculations.

In conclusion, we have presented a method that allows binding energies of negative ions to be determined with uncertainties an order of magnitude smaller than in previous experiments. The method was applied to study the electron affinity of atomic oxygen to be

1.461,112,972(87) eV. In addition, we measured the fine-structure splitting of the ground state, yielding the result 21.955,44(30) meV. The accuracy reached for both measured values is unprecedented. The method is generally applicable and can be used to measure electron affinities of any atomic system with bound excited states, which constitute a majority of the elements in the periodic table. Essentially all atomic negative ions can be produced in a cesium sputter ion source, so the current limitation for the method is the accessibility of narrow bandwidth tunable lasers, in particular in the infrared wavelength region. The method is also applicable for determinations of detachment energies for a large number of molecules. These high-precision measurements are expected to inspire new interest in theoretical work within the field of atomic many-body theory. In particular, the method will enable investigations of isotope effects on the electron affinity at a level of detail never before achieved. We are now in the process of applying the technique to study the isotope shift of the electron affinity between $^{16}$O and $^{18}$O. Detailed investigations of the isotope shift are of interest in both atomic physics and nuclear physics since the isotope shift can be used to probe both electron correlation and nuclear charge distributions.

## Methods
### Experimental setup
The experiment was performed at the DESIREE facility, which consists of two storage rings placed inside a vacuum chamber kept at 13 K. Here, we use only one of the rings, as shown in Fig. 2. The low temperature provides very efficient cryogenic pumping contributing to a low particle density of less than $10^4$ molecules per cm³. At 13 K this corresponds to less than $2 \times 10^{-14}$ mbar. The negative oxygen ions are formed in a cesium sputter ion source (SNICSII[53]) with a SnO cathode. A sputtering voltage of 5.8 kV is used. The ions are extracted from the source and accelerated to form a beam with 30 keV kinetic energy. A 90°-bending magnet with a mass resolution m/Δm (in combination with slits of adjustable sizes) of more than 200 is used for mass selection, after which the beam is bunched and injected into one of the storage rings. The length of the ion bunch is about 13 μs, this corresponds to one revolution in the ion ring and, thus, the entire ring is almost completely filled with ions during one injection. The ion beam current measured before the injection is ~5.5 nA. The timing scheme of a measurement cycle is shown in Fig. 6a and the optical setup is shown in Fig. 6b. The start of a measurement cycle is initiated by turning the detector to a low operation mode by reducing the voltage on the glass plate and front of the microchannel plate (MCP) of the detector to zero. This is done to prevent saturation of the detector from the high rate of neutral atoms during the laser depletion of the upper fine structure level. It takes a few seconds to ramp down the detector, and when that is done, the ions are injected into the storage ring where a laser beam of very high power, ~2.5 W, and photon energy which is 170 μeV below the electron affinity, but several meV above the electron detachment threshold for the upper fine structure level, is applied (arrow set "a" in Fig. 1). The laser co-propagates with the ion beam. Within a few seconds, this high-power laser almost completely empties the ion beam of ions in the upper fine-structure level through photodetachment. After 20 s of depletion, the power of the laser is decreased by a factor of 10–40 and the detector voltages are restored to their operation values. The photon energy is changed, corresponding to energy just below the photodetachment threshold of the ground state; this will be the center of the scan. The scan range used is 20 GHz (about 80 μeV) and the scan time is 30 s. Four photon energy scans are performed before the ions are dumped in a Faraday cup as a control measurement of how many ions are left in the ring. The full measurement cycle is 165 s. After one measurement cycle, a mirror is inserted by a motorized controller in the laser beam path, changing the laser to be counter-propagating with the ion beam. A new measurement cycle is started where the

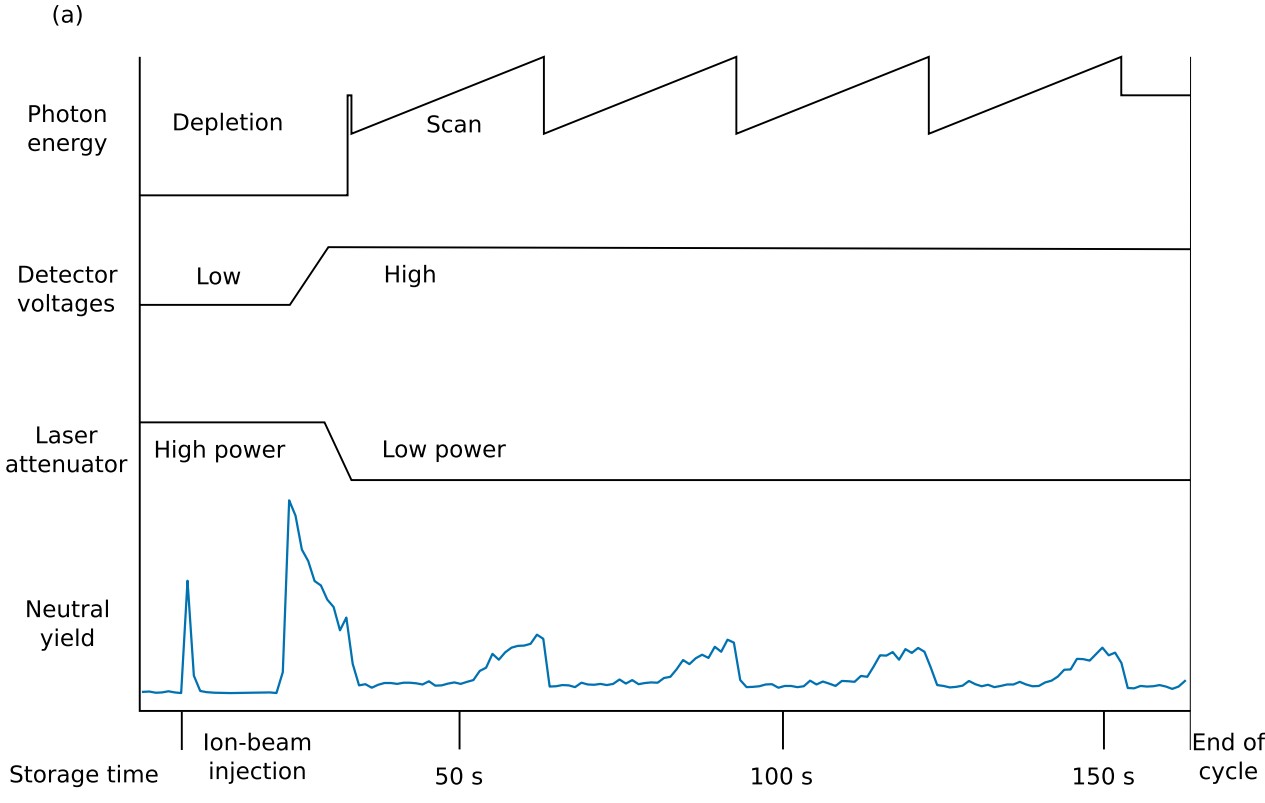

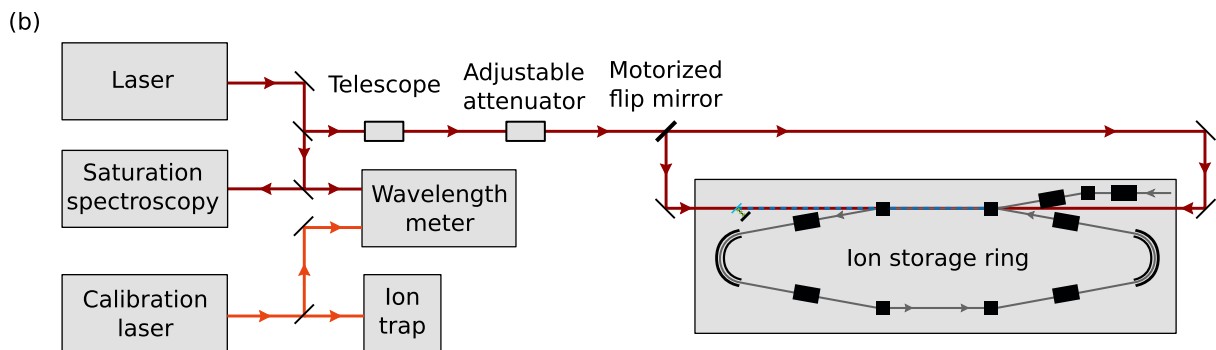

**Fig. 6 | Timing and optical scheme of the measurement procedure.** In **a** the timing of the laser, attenuator, and detector is shown together with a real-time example of the neutral yield on the detector. Note that there is still a signal even though the detector is in a low operation mode for the first ~30 s. This is due to the high number of neutrals generated by the depletion of the ions in the excited state. In **b** the optical setup with the laser, calibration laser, wavelength meter, attenuator, flip mirror, and overlap with the ion beam is shown. A small fraction of the laser light used for photodetachment is picked off by a glass plate in the beam path and guided to the wavelength meter to measure the photon energy continuously during the measurement cycle.

same procedure is performed with the only difference being in the photon energy, due to the opposite Doppler shifts of the two light-propagation directions. This procedure is necessary as the uncertainty in the Doppler shift from the precision with which the energy of the ion beam is known would otherwise be the dominating source of error. The measurement procedure is repeated several times.

**Laser and photon energy measurement**
The laser is a SolsTiS narrow-linewidth titanium-sapphire laser from M Squared[54]. It generates tunable light in the 700–1000 nm range with a maximum output power of 5 W and a linewidth smaller than 50 kHz. The laser photon energy is measured using a HighFinesse WS8-2 wavemeter specified with an absolute accuracy of 10 MHz (0.041 μeV) when using a calibration photon energy further away than about 3.4 meV from the measured photon energy. To achieve this high accuracy, the wavemeter is calibrated using light from a Toptica TA pro diode laser locked to the dipole-forbidden transition $^5S_{1/2}$-$^4D_{5/2}$ of an $^{88}Sr^+$ ion. This transition has a frequency of 444.779,044,095,484,6(15) THz and is known to the Hz level[55]. This, combined with the kHz linewidth of the stabilized laser, is more than enough to calibrate the wavemeter.

The calibration frequency is about 90 THz (0.37 eV, 216 nm) away from the photon energy of the light used for the experiment. Therefore, we made a control measurement of the calibration accuracy using saturation spectroscopy of cesium. The transition between the $^2S_{1/2}$-$^2P_{3/2}$ levels is well known and the hyperfine splittings of these levels are even more accurately known. The hyperfine splittings are observed using saturation spectroscopy and a shift of +0.029 μeV is found. This shift is subtracted from the measured photon energies. The uncertainty from the wavemeter does, however remain 10 MHz (0.041 μeV)

since the manufacturer specifies an uncertainty of 10 MHz if the calibration source is further away than 2 nm from the measured values.

## Measurement details

A measurement of the ratio of ions in the ground state relative to the excited state is done by injecting and storing 7.8 nA of ions in the ring. After 40 s, the beam is dumped in the Faraday cup and the current, $I_0$, is recorded and averaged over ten measurement cycles. A second measurement is done with the laser tuned to deplete the excited state with a laser power of 2.5 W and the ion beam current $I_{ex}$ is measured. Again, an average of over ten measurement cycles is recorded. The ratio between the measured ion beam currents $I_0/I_{ex}$ is 6.6(3)/4.1(2); in other words, assuming that all ions in the excited state are depleted using the laser, 38(6)% of the ions are in the excited state. This is consistent with a population determined by the degeneracy of the state, where we would expect one-third of the ions to be in the excited $J = 1/2$ level.

The laser power is measured after the laser beam propagates through a telescope and adjustable attenuator, which are located about 9 m before the window for the co-propagating direction and 4 m before the entrance window for the counter-propagating direction. The difference in laser beam paths causes slightly more mirror losses in the co-propagating direction, but this is largely compensated by the fact that the laser beam travels through the detector glass plate before interacting with the ion beam in the counter-propagating alignment. The resulting powers in both directions are hence similar.

An important aspect of the precision of the experiment is the energy distribution of the ions in the beam. The distribution of the revolution frequencies of the individual ions in the ring, known as the Schottky spectrum[56], is measured by a pickup electrode, an amplifier, and a spectrum analyzer. When storing the ion beam, the Schottky signal is used to monitor the revolution frequency of the ions. The spread of the frequency can be converted to a momentum spread and the beam energy spread is obtained. The energy spread obtained from the Schottky spectrum is only used as a beam diagnostics tool and does not give an accurate measurement of the absolute energy spread. In the fits to obtain the threshold values, the energy spread is one of the fit parameters in the Gaussian function as described in Eqn. (3).

The energy spread has several possible sources. The energy of the ions will acquire a spread when accelerated out of the ion source. In addition, when the ions are stored in the ring, the width of the energy distribution of the ions increases further due to intrabeam scattering. The spread increases slightly with storage time and typically stabilizes after about 30 s of storage. From our experience of running experiments at DESIREE, the ion beam energy spread is typically on the order of 0.1% of the total beam energy, although the energy spread is a bit smaller for higher beam energies. Attempts have been made to minimize the energy spread by using a gas discharge ion source instead of the sputtering source. This, however, results in a very small improvement since the dominating contribution to the energy spread comes from the intrabeam scattering while storing the ions. The scattering depends on the ion density and the velocity of the ions. The gas discharge source performed poorly considering the stability of the ion beam current, and therefore the sputter source was chosen. The beam current is thus selected to be as low as possible while still getting a significant photodetachment signal. The laser scanning power also affects the energy spread of the ions when scanning close to the threshold for photodetachment. This effect is, as discussed previously, removed by selecting a low laser power while scanning.

Any additional uncertainties to the final electron affinity beyond those already discussed are considered to be negligible compared to the final uncertainty of 21 MHz. The maximum effect from a nonzero

angle between the laser and ion beam is estimated to be less than 1 MHz, and the effect from the divergence of the laser beam is even smaller.

## Data availability

The data and data analysis code related to this paper is available from the authors upon request.

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

## Acknowledgements

We would like to thank Fabian Pokorny, Marion Mallweger, Harry Parke, and Markus Hennrich from the Trapped Ion Quantum Technologies group for their contribution to providing the wavemeter calibration signal. This work was performed at the Swedish National Infrastructure, DESIREE (Swedish Research Council Contract Nos. 2017-00621 and 2021-00155). Furthermore, H.C., H.Z., H.T.S., D.H., and J.G. thank the Swedish Research Council for individual project grants (with contract Nos. 2019-04379, 2020-03437, 2018-04092, 2016-03650, 2020-03505, and 2020-05467).

## Author contributions

M.K.K., K.C., G.E., H.T.S., and D.H. planned and developed the experiment. M.K.K., K.C., G.E., O.M.H., E.K.A., N.d.R., M.K, N.P., J.E.N.-N., and S.S. performed the experiment and assisted with taking the data. M.K.K., K.C., and O.M.H. set up and operated the laser system. A.S., M.B., S.R., P.R., M.B., A.K., and J.D.A. set up and operated DESIREE. M.K.K. and G.E. analyzed the data. M.K.K., K.C., J.G., H.T.S., and D.H. wrote the manuscript draft. H.C., H.Z., H.T.S., and D.H. supervised the participants. All authors contributed to the discussion of the manuscript.

## Funding

## Competing interests

The authors declare no competing interests.
