## [Peer Review File · Nature Communications]

High-precision electron affinity of oxygenREVIEWER COMMENTS

Reviewer #1 (Remarks to the Author):

This paper reports the measurement of the electron affinity of both fine structure states of the atomic oxygen anion. The results for the ground states is in disagreement with previous work in 1985, but it agrees with studies from 2003 and 2010 (see Fig 5 of the manuscript). The estimated error bar in the current work, about 0.1 eV, is an order of magnitude better than previous work. This makes the experimental value substantially more accurate than present theoretical values, thus posing a challenge for theoretical work.

One isotope of oxygen has been measured, but the technique that was used is claimed to be quite general, and in ongoing work two isotopes of oxygen are being studied. Currently, the difference between theory and experiment for the isotope effect is within the experimental error bar, but the present precision would also pose a challenge to theory for the isotope effect. The isotope effect is important for the study of nuclear structure, and as a test on electron correlation effects in theoretical work.

The paper is well written and the importance of electron affinity for a wide range of topics in atomic, molecular and nuclear physics is summarized.

The promise of ongoing work on isotope effects, and the possibility to study other atomic and even molecular ions builds suspense, but should in no way hold back publication of this groundbreaking work. I believe that much of the progress in atomic and molecular physics builds on gains in spectroscopic resolution.

I have two suggestions for the present manuscript: although the method is described in detail, it may not be obvious to the wider audience what is the key advancement that makes the higher precision possible. My other question is triggered by the claim that the method is general: what are the main issues/challenges when extending this work to other species?

Reviewer #2 (Remarks to the Author):

Title: High-precision electron affinity of oxygen
Manuscript#: NCOMMS-22-18532-T

Reviewer: John N. Yukich

Overall comments to authors and editors: This manuscript represents an unusually large improvement in laser photodetachment spectroscopy of negative ions. The authors argue, and correctly so, that negative ion physics has wide applicability and interest in other areas of science. The authors provide a suitable background and motivation section necessary for the wide audience of Nature

communications. They also provide a good description of the novel technique that permits their precision measurements.

Key results and significance: The major advancement in this area is the authors' ability to preferentially select or purify the initial state of an ensemble of ions in a cold ion storage ring, which largely eliminates background photodetachment signal. This effect combined with a narrow-linewidth laser and a precision wavelength meter yields a ten-fold increase in spectroscopic precision, allowing for new measurements of electron affinities of negative ions, with unprecedented precision.

Originality and reproducibility: The authors have gone to extensive efforts to diminish the uncertainty on their photon energy measurements using a high-precision, carefully-calibrated wavelength meter. Their results include a measurement of the electron affinity of O with an uncertainty an order of magnitude lower than the previous best results. Their value for this electron affinity is also in numerical agreement with two prior measurements using different techniques. The authors also include a careful, extensive analysis of all the numerical uncertainties contributing to their final values. Given the care shown in the data analysis, I am confident that the results are reproducible, although I don't think there are many research groups equipped to do comparable work.

Scientific concern: My only significant scientific concern is with regard to the below-threshold photodetachment shown in Fig. 3. These are significant background signals. I strongly recommend that the authors address this directly, relate it to the C term in Eq. 2, and comment on its source. Is it possibly brought about by collisional redistribution of the initial states during the state-selection process? It seems unclear at the moment (see further comments in detailed section below).

Significance: This experimental work raises the bar on precision photodetachment spectroscopy by an order of magnitude, an improvement that is uncommon today. The enhanced precision now available with the authors' technique will challenge the prevailing theoretical methods of atomic structure theory, ultimately yielding deeper investigations and better understanding of the electron correlation effects that play such a large role in physics and chemistry. Given the ubiquitous importance of oxygen, this particular manuscript will be of general interest to scientists in several areas: atomic physics, physical chemistry, atmospheric science, etc. The huge methodological improvement will be of interest to numerous experimentalists.

Overall recommendation to the editors: This work represented by this manuscript is definitely worthy of publication in Nature. I have studied the manuscript in detail and provide below a large number of comments, observations, and recommendations for improving clarity of the paper, which I believe will also increase its already high credibility. I highly recommend publication after minor adjustments outlined below.

Thank you for the opportunity to assist with review of this wonderful work! I will be more than happy to review subsequent versions. I strongly believe the manuscript can be easily revised.

- Sincerely,

John Yukich

Detailed Comments by line and section:

1. Lines 28-30: It would be helpful to mention here that the lack of optically allowed transitions also limits prospects for laser cooling.

2. Line 33: "we are able to eliminate the background" seems like a very strong statement given that the data in Fig. 4 indicate a baseline detachment on the order of 20% of the highest count rate shown. Presumably this background is given by the term C in Equation 2. Adjusting the claim to "nearly eliminate" or "largely eliminate" would seem more appropriate for the data.

3. Line 42: recommend changing to more conventional wording of “molecular negative ions” or “molecular anions”.
4. Lines 49-50: this mention of recent work in laser cooling should include Ref. 15 and Ref. 16.
5. Lines 55-56: Mention of “many chemical properties” that can be determined from the EA should include a couple of examples with references – important for the broad audience of Nature.
6. Lines 59-63: Although I have 30+ years’ experience working with negative ion spectroscopy, I initially found the wording here to be confusing. The text refers to the attractive Coulomb potential felt by a valence electron in a negative ion, but then essentially says that the same valence electron experiences little or no Coulomb attraction. Thus, the wording sounds contradictory. Recommendation: adjust the second sentence to something like this: “These valence electrons thus experience little or no long-range net force.”
7. Line 64: mention of the $1/r^4$ polarization potential could benefit from a reference.
8. Lines 78-79 mentions that the electron affinity is generally the only high-precision measurement that can be made. In fact, however, other electronic transitions can be measured in addition to the electron affinity transition – as you mention later in lines 111-112.
9. Lines 107-109: just an observation here: back in Dan Larson’s group, we measured and published the electron affinity of OH. I’ve found that we can also make OD⁻ in our Penning ion trap, and I hope one day to measure its EA and compare!
10. Line 120: it would help to explain briefly why $l = 0$ for oxygen, that s-wave detachment dominates at low photon energies above threshold.
11. Lines 128-129: I’m pretty sure that what really determines the excited state populations is a comparison of the ion production energies, not with the electron affinity, but with the spin-orbit splitting.
12. Line 129: “typical . . . typically” sounds redundant.
13. Lines 137-147: I wonder how your system reduces the chance of collisional redistribution of initial states while the ions are circulating in the storage ring. Obviously, you have a good vacuum, but the ions are moving quite fast, too, right? . . . is redistribution somehow not a problem? What is the average speed of the ions in the ring? This seems like an important topic to address, especially given the below-threshold background signals shown in Fig. 3.
14. Line 152: the transition between these two sentence topics feels extremely abrupt. Another sentence or two about the significance of the 13 K temperature would help here.
15. Line 156: it would help to specify parallel and anti-parallel to the ion beam.
16. Figure 1: the slanted hatch marks at the top of the 3P2 level remind me of symbols commonly used to indicate an energy continuum. The electron is certainly detached into a continuum, but the ground state of O is definitely a discrete state and itself does not border a continuum. This is a source of possible confusion, but it could be easily addressed in the caption.
17. Figure 1 caption: for consistent wording, adjust last sentence to read: “. . . photon energy needed for photodetachment from the excited state . . . photon energy needed to photodetach from both the ground state and excited state.”
18. Lines 170-172: it would be helpful to mention that C represents possible non-zero background

detachment. This would be a good place to mention that C is greatly diminished with the state-selective process of preparing the ion beam. This leads to a question: what would C be without your state-selective preparation process? This seems to be a key element of your enormous enhancement in precision.

19. Figure 2: excellent, very well done!

20. Figure 3: the symbol for the ion used at the top of the antiparallel data matches the symbol used in Fig. 2. However, the ion symbol used at the top of the parallel data has a square in the middle of it and does not match the other ion symbols. I can't tell if this is deliberate, but I assume they should all match.

21. Section 3: I really appreciate the care given here to the uncertainty budget!

22. How do we explain the non-zero background detachment signals shown in Fig. 3?

23. Lines 219-221: It's unclear why a smaller number of scans were collected when measuring the upper fine-structure level. Given that the cross section is smaller for this transition and the excited state population is lower than that of the ground state, it seems you would want to collect a larger number of scans rather than a smaller number of scans.

24. Line 283: there are some extraneous words here: "agrees with to" and "value of or".

25. Section 5 (lines 290 and 294): refer to accuracy. Your novel method has improved by an order of magnitude the precision of the binding energy measurements. This is an important distinction from accuracy. A measurement can be highly precise but demonstrably inaccurate due to systematic error. Lines 296-297, however, correctly refer to high precision.

26. Line 300: should be adjusted to read "We are now in the process of applying the new technique . . ."

27. Section 6.1: What is the background pressure in the vacuum chamber? I ask this because I am still wondering about the possibility of collisional redistribution of the initial ion state due to collisions. Lines 410-411 refer to energy spread arising from intrabeam scattering during the storage, which makes me think there must be significant collisional redistribution. Is this why the baseline detachment in Fig. 3 is non-zero?

28. It would also help to clarify: are the ions continuously injected into the storage ring, or are they injected in sample bunches?

29. Line 314: perhaps add one word to read ". . . bending magnet with a mass resolution of ..."

Reviewer replies

Reviewer #1:

I have two suggestions for the present manuscript: although the method is described in detail, it may not be obvious to the wider audience what is the key advancement that makes the higher precision possible.

We have added a few sentences after the first sentence in subsection “Precision Measurements” to more clearly explain what made the high precision possible. We point out that it is the dramatic reduction of the background below the threshold that is the key advancement that made the high precision possible.

My other question is triggered by the claim that the method is general: what are the main issues/challenges when extending this work to other species?

It was already pointed out in the conclusions that the method is applicable to essentially all atomic negative ions. We have now added that the method is applicable to all elements with bound metastable states, which constitutes a majority of the elements in the periodic table. We also state that almost all negative ions can be produced in a cesium sputter source and that the limiting factor therefore is the accessibility to narrow bandwidth lasers particularly in the infrared wavelength region. This information is now added to the conclusion.

Reviewer #2:

Scientific concern: My only significant scientific concern is with regard to the below-threshold photodetachment shown in Fig. 3. These are significant background signals. I strongly recommend that the authors address this directly, relate it to the C term in Eq. 2, and comment on its source. Is it possibly brought about by collisional redistribution of the initial states during the state-selection process? It seems unclear at the moment (see further comments in detailed section below).

A more careful discussion about the level of the signal below the threshold has been added in the beginning of the subsection “precision measurements”. : “The signal below the threshold (the level of which corresponds to the constant, C , in Eq. 2) is a combination of detector background, signal from atoms created by collisions with the residual gas, and a small contribution from photodetachment of ions remaining in the excited state. Without depletion of ions in the excited state, the background signal would have been more than 20 times larger, and would thus have prevented an accurate threshold determination.”

The possibility of collisional redistribution has been experimentally investigated and is shown to give a negligible contribution to the signal below the threshold. This can be seen by the level of the signal below the threshold found in Figure 6 where, if redistribution would take place, the level of the signal below the threshold would increase towards the end of the measurement cycle. To save space, this information is not presented in the paper.

Detailed Comments by line and section:

1. Lines 28-30: It would be helpful to mention here that the lack of optically allowed transitions also limits prospects for laser cooling.

We had this in our original version, but it was deleted due the word limit of the abstract. However, the prospect of laser cooling of negative ions is mentioned in the introduction.

2. Line 33: “we are able to eliminate the background” seems like a very strong statement given that the data in Fig. 4 indicate a baseline detachment on the order of 20% of the highest count rate shown. Presumably this background is given by the term C in Equation 2. Adjusting the claim to “nearly eliminate” or “largely eliminate” would seem more appropriate for the data.

“eliminate” has been changed to “strongly reduce”

3. Line 42: recommend changing to more conventional wording of “molecular negative ions” or “molecular anions”.

“negative molecular ions” has been changed to “molecular negative ions”

4. Lines 49-50: this mention of recent work in laser cooling should include Ref. 15 and Ref. 16.

The suggested references have been included.

5. Lines 55-56: Mention of “many chemical properties” that can be determined from the EA should include a couple of examples with references – important for the broad audience of Nature.

We added “e.g. the electronegativity or the electrophilicity index” to the sentence and a few relevant references.

6. Lines 59-63: Although I have 30+ years’ experience working with negative ion spectroscopy, I initially found the wording here to be confusing. The text refers to the attractive Coulomb potential felt by a valence electron in a negative ion, but then essentially says that the same valence electron experiences little or no Coulomb attraction. Thus, the wording sounds contradictory. Recommendation: adjust the second sentence to something like this: “These valence electrons thus experience little or no long-range net force.”

The sentence "These valence electrons thus experience no, or very little, long-range Coulomb force." has been changed to "These valence electrons thus experience little or no long-range net force."

7. Line 64: mention of the $1/r^4$ polarization potential could benefit from a reference.

A reference has been added.

8. Lines 78-79 mentions that the electron affinity is generally the only high-precision measurement that can be made. In fact, however, other electronic transitions can be measured in addition to the electron affinity transition – as you mention later in lines 111-112.

We agree that this sentence is not correct. We have changed the sentence "Therefore, the electron affinity is generally the only property of which high-precision measurements can be performed."

to

"The most characteristic general property of negative ions is the electron affinity of the corresponding neutral atom or molecule."

9. Lines 107-109: just an observation here: back in Dan Larson's group, we measured and published the electron affinity of OH. I've found that we can also make OD- in our Penning ion trap, and I hope one day to measure its EA and compare!

Such an experiment would be very interesting.

No changes to manuscript.

10. Line 120: it would help to explain briefly why $l = 0$ for oxygen, that s-wave detachment dominates at low photon energies above threshold.

The sentence

"In the case of oxygen, $l = 0$ yields a sharp onset at threshold, which can be determined with high precision."

has been changed to

"In the case of oxygen the 2p electron can be emitted as an s or d electron. Close to the threshold the s-wave dominates due to the centrifugal barrier for the d-wave, yielding a sharp onset ($l=0$ in Eq. 1 for the s-wave) that can be determined with high precision."

In addition, two relevant references have also been added.

11. Lines 128-129: I'm pretty sure that what really determines the excited state populations is a comparison of the ion production energies, not with the electron affinity, but with the spin-orbit splitting.

"since the energies involved in ion production are much larger than the typical electron affinities which typically are of the order of 1 eV"

has been changed to

“since the energies involved in ion production are in general much larger than the energy splittings between the levels involved.”

12. Line 129: “typical . . . typically” sounds redundant.

We now avoid the redundant use of “typically”.

13. Lines 137-147: I wonder how your system reduces the chance of collisional redistribution of initial states while the ions are circulating in the storage ring. Obviously, you have a good vacuum, but the ions are moving quite fast, too, right? . . . is redistribution somehow not a problem? What is the average speed of the ions in the ring? This seems like an important topic to address, especially given the below-threshold background signals shown in Fig. 3.

This is discussed under the point scientific concern (the first point raised by reviewer 2). Additional information has been added in the beginning of subsection “precision measurements”

14. Line 152: the transition between these two sentence topics feels extremely abrupt. Another sentence or two about the significance of the 13 K temperature would help here.

The sentence “The current experiment utilizes a storage ring held at 13 K and a high power laser to produce an ion beam almost completely in the ground state thus avoiding photodetachment from the excited state that would otherwise affect the measurements.” has been added.

15. Line 156: it would help to specify parallel and anti-parallel to the ion beam.

We changed the text to “laser beams either co- or counter-propagating with respect to the ion beam.”

16. Figure 1: the slanted hatch marks at the top of the 3P2 level remind me of symbols commonly used to indicate an energy continuum. The electron is certainly detached into a continuum, but the ground state of O is definitely a discrete state and itself does not border a continuum. This is a source of possible confusion, but it could be easily addressed in the caption.

We have added the sentence

“The hatched area marks the continuum of an oxygen atom in its ground state and a free electron.”

17. Figure 1 caption: for consistent wording, adjust last sentence to read: “. . . photon energy needed for photodetachment from the excited state . . . photon energy needed to photodetach from both the ground state and excited state.”

This has been corrected.

18. Lines 170-172: it would be helpful to mention that C represents possible non-zero background detachment. This would be a good place to mention that C is greatly diminished with the state-selective process of preparing the ion beam. This leads to a question: what would C be without your state-selective preparation process? This seems to be a key element of your enormous enhancement in precision.

We have responded to this question under the point “scientific concern”. Additional information has been added in the beginning of subsection “Precision measurements”. Further, we added a sentence below Eq. 2 to explain the importance of reducing the background.

19. Figure 2: excellent, very well done!

Thank you!

20. Figure 3: the symbol for the ion used at the top of the antiparallel data matches the symbol used in Fig. 2. However, the ion symbol used at the top of the parallel data has a square in the middle of it and does not match the other ion symbols. I can’t tell if this is deliberate, but I assume they should all match.

Thanks, this was a mistake and has now been corrected.

21. Section 3: I really appreciate the care given here to the uncertainty budget!

Thank you!

22. How do we explain the non-zero background detachment signals shown in Fig. 3?

We have responded to this question under the point “scientific concern”. Additional information has been added in the beginning of subsection “Precision measurements”.

23. Lines 219-221: It’s unclear why a smaller number of scans were collected when measuring the upper fine-structure level. Given that the cross section is smaller for this transition and the excited state population is lower than that of the ground state, it seems you would want to collect a larger number of scans rather than a smaller number of scans.

We had a limited beamtime and had to prioritize during the experiment. The electron affinity measurement is the main topic of this paper and was therefore prioritized.

24. Line 283: there are some extraneous words here: “agrees with to” and “value of or”.

This is changed to “agrees with” and “value of”

25. Section 5 (lines 290 and 294): refer to accuracy. Your novel method has improved by an order of magnitude the precision of the binding energy measurements. This is an important distinction from accuracy. A measurement can be highly precise but demonstrably inaccurate due to systematic error. Lines 296-297, however, correctly refer to high precision.

The sentence “We have presented a method that allows binding energies of negative ions to be determined with an accuracy an order of magnitude higher than in previous experiments.” is changed to “... determined with uncertainties an order of magnitude smaller than in ...” to avoid this mixup between accuracy and precision.

26. Line 300: should be adjusted to read “We are now in the process of applying the new technique . . .”

The corresponding change has been made.

27. Section 6.1: What is the background pressure in the vacuum chamber? I ask this because I am still wondering about the possibility of collisional redistribution of the initial ion state due to collisions. Lines 410-411 refer to energy spread arising from intrabeam scattering during the storage, which makes me think there must be significant collisional redistribution. Is this why the baseline detachment in Fig. 3 is non-zero?

It is stated in section 6.1 that particle density is 10^4 molecules per cm^3 . At 13 K this corresponds to less than 2×10^{-14} mbar. This information is now added to the manuscript.

We have responded to this question about collisional redistribution under the point “scientific concern”. Additional information has been in the beginning of subsection “Precision measurements”

28. It would also help to clarify: are the ions continuously injected into the storage ring, or are they injected in sample bunches?

We added information about this in the methods section:

“... after which the beam is bunched and injected into one of the storage rings. The length of the ion bunch is about 13 μs , corresponding to one revolution in the ion ring and, thus, the entire ring is almost completely filled with ions during one injection.”

29. Line 314: perhaps add one word to read “... bending magnet with a mass resolution of ...”

The corresponding change has been made.

REVIEWERS' COMMENTS

Reviewer #2 (Remarks to the Author):

Title: High-precision electron affinity of oxygen

Manuscript#: NCOMMS-22-18532-T

Reviewer: John N. Yukich

Comments to authors and editors following revision of manuscript: Thank you again for the opportunity to review this manuscript. Overall, the authors have done an excellent job of responding to both my comments and suggestions as well as those of the other reviewer. The adjustments made by the authors have greatly improved the paper, both in its scientific attributes and in its clarity. I strongly recommend it now for publication in Nature Communications.

- Sincerely, John N. Yukich

Dear reviewers,

We thank you for your work and much appreciate the feedback. We find that the article has much improved when taking your comments into consideration.

Moa K Kristiansson